An experimental analysis of graph representation learning for Gene Ontology based protein function prediction

http://orcid.org/0009-0003-0640-4364 Vu Thi Thuy Duong 1
Kim Jeongho 2
Jung Jaehee 2 jhjung@mju.ac.kr
1 Faculty of Fundamental Sciences, University of Medicine and Pharmacy at Ho Chi Minh City , Ho Chi Minh City , Vietnam
2 Department of Information and Communication Engineering, Myongji University , Yongin , Republic of South Korea
Uversky Vladimir
Electronic publication date: 2024 Nov 14
Publication date: 2024
Volume: 12
Electronic Location ID: e18509
Received 2024 Sep 5; Accepted 2024 Oct 21
Copyright: © 2024 Vu et al.
Copyright year: 2024
Copyright holder: Vu et al.
License: This is an open access article distributed under the terms of the Creative Commons Attribution License, which permits unrestricted use, distribution, reproduction and adaptation in any medium and for any purpose provided that it is properly attributed. For attribution, the original author(s), title, publication source (PeerJ) and either DOI or URL of the article must be cited.
License URL: https://creativecommons.org/licenses/by/4.0/

Keywords: Protein function prediction, Deep learning, Graph representation learning, Gene Ontology

Funding: National Research Foundation of Korea (NRF) Korea government (MSIT) NRF-2022R1F1A1061476 This work was supported by the National Research Foundation of Korea (NRF) grant funded by the Korea government (MSIT) (NRF-2022R1F1A1061476). The funders had no role in study design, data collection and analysis, decision to publish, or preparation of the manuscript.

==============================
Understanding protein function is crucial for deciphering biological systems and facilitating various biomedical applications. Computational methods for predicting Gene Ontology functions of proteins emerged in the 2000s to bridge the gap between the number of annotated proteins and the rapidly growing number of newly discovered amino acid sequences. Recently, there has been a surge in studies applying graph representation learning techniques to biological networks to enhance protein function prediction tools. In this review, we provide fundamental concepts in graph embedding algorithms. This study described graph representation learning methods for protein function prediction based on four principal data categories, namely PPI network, protein structure, Gene Ontology graph, and integrated graph. The commonly used approaches for each category were summarized and diagrammed, with the specific results of each method explained in detail. Finally, existing limitations and potential solutions were discussed, and directions for future research within the protein research community were suggested.

Introduction

Protein function annotation is one of the most fundamental research topics in bioinformatics. Proteins are known as the building blocks of life, as they are responsible for a diverse range of activities in living organisms, such as catalyzing biochemical reactions, providing cellular structures, transporting nutrients, and regulating processes like gene expression and signal transduction. The identification of protein roles is not only crucial for biological systems, but can also enhance other aspects, such as drug discovery, disease therapies, agriculture or manufacturing. However, current manual protein annotation performed by experts is costly and time-consuming, which could not keep up with the huge number of new proteins generated from high-throughput sequencing techniques. Specifically, there are more than 249 million unreviewed proteins in the UniProtKB database (The UniProt Consortium, 2023), only around 570 thousand sequences are manually annotated until April 2024. Thus, the development of accurate and effective computational predictors for protein function prediction (PFP) is imperative to bridge this gap.

There are several standardized schemes describing protein functions, such as the Functional Catalogue (FunCat) (Ruepp et al., 2004), Kyoto Encyclopedia of Genes and Genomes (KEGG) (Kanehisa et al., 2023) and Gene Ontology (GO) (Ashburner et al., 2000; Consortium et al., 2023). At present, the GO database is the most widely system used for protein functional annotation. The GO knowledgebase describes functions of genes and gene products in three domains: molecular function (MF), biological process (BP), cellular component (CC). GO terms (also known as GO classes) are organized in a hierarchical directed acyclic graph (DAG), where an edge denotes a specific parent-child relationship between two GO terms, such as is a, part of, has part, regulates. Thus, protein annotation is propagated by a principle called true path rule (Valentini, 2010), in which if a protein is associated with a GO class, the protein is annotated with all of that GO term’s parents.

The research community has witnessed several computational methods for annotating GO functions for proteins over the last few decades (Makrodimitris, Van Ham & Reinders, 2020; Vu & Jung, 2021), ranging from conventional to machine learning based solutions. Homology based function annotation marked the early stages of automated protein function prediction. This approach assumes that proteins with similar sequences are likely to perform similar roles. In a typical workflow, a query protein sequence is compared against a database of annotated proteins. GO terms associated with the top sequence matches are scored and selected for transfer to the query protein. Various studies following this approach have utilized different sequence search tools, such as BLAST (Altschul et al., 1997), PSI-BLAST (Altschul et al., 1997), and DIAMOND (Buchfink, Xie & Huson, 2015), and have proposed multiple ways for scoring GO terms (Martin, Berriman & Barton, 2004; Hawkins et al., 2009; Gong, Ning & Tian, 2016). Determining protein characteristics through sequence homology is straightforward, but it has drawbacks that can lead to inaccurate function transfer. For example, there are proteins with similar amino acid sequences that may perform different functions, while others with little sequence similarity might share the same biological roles (Sasson, Kaplan & Linial, 2006). Advances in machine learning have opened new avenues for computational PFP. In this context, protein function annotation is framed as a multi-label classification problem, as a single protein can exhibit more than one function. Studies employing machine learning models, such as logistic regression, support vector machines (SVM), and k-nearest neighbors ( k-NN), often rely on extensive feature engineering from several sources to extract relevant protein features (Lobley et al., 2008). Also, predictions are typically generated by component classifiers, which are then ensembled to ultimately assign GO terms to proteins (You et al., 2018, 2019). It is noteworthy that homology based function annotation is still extensively utilized, particularly as an important component in lately proposed methods (Zhang & Freddolino, 2024).

Recent studies have increasingly focused on deep learning based models that leverage vast amounts of biological data, computational resources, and sophisticated algorithms designed to automatically capture features from raw data (Bonetta & Valentino, 2020; Dhanuka, Singh & Tripathi, 2023; Yan et al., 2023). In terms of input data, there have been two main approaches. On one hand, researchers focus on inferring associated functions using only protein sequences (Cao et al., 2017; Sureyya Rifaioglu et al., 2019; Nauman et al., 2019; Kulmanov & Hoehndorf, 2020; Wu et al., 2023), as this is the only information available for all proteins. On the other hand, integrating multiple biological information sources has become a prominent research direction, and this approach yields the best performance in the latest Critical Assessment of protein Function Annotation, CAFA5 (Zhou et al., 2019b; Friedberg et al., 2023). In addition to learning protein sequence representations (Cui, Zhang & Zou, 2021) using large protein language models (Elnaggar et al., 2022), deep graph learning models have recently gained popularity for capturing complex relationships within diverse biological networks, including protein-protein interaction (PPI) network, protein structures, the GO tree, and protein annotations, thereby enhancing the inference of protein characteristics (You et al., 2021; Lai & Xu, 2022).

It is necessary to conduct a systematic review of the advances in graph representation learning strategies for protein function prediction. Several surveys summarized a few years ago (Makrodimitris, Van Ham & Reinders, 2020; Bonetta & Valentino, 2020) may not reflect the latest developments in the field. The most recent reviews we found are from Dhanuka, Singh & Tripathi (2023) and Yan et al. (2023), but they primarily focus on protein sequence based methods and machine learning models, respectively. In this study, we review state-of-the-art applications of graph embedding for protein function prediction that have emerged in the last 5 years. In addition, we have written about recent protein function prediction research trends at url: https://zenodo.org/records/13828158 to keep researchers updated. This review aims to provide researchers with a comprehensive overview, highlighting major research directions, commonly used techniques, and specific improvements, organized by different types of biological graph data. Additionally, we discuss the open challenges and potential solutions, offering perspectives for further developments in the functional proteomics field.

Survey methodology

We created a preliminary list of articles using Google Scholar and PubMed. The search queries were combinations of the following keywords: “protein function prediction”, “protein function annotation”, “Gene Ontology”, “graph neural network”, “graph representation learning”, “graph embedding”, and “deep learning”. To focus on recent studies, we filtered out results published before 2019.

Initially, articles were selected based on their titles and abstracts. Subsequently, full-text articles were assessed to determine whether they qualify as a significant method for Gene Ontology based protein function prediction using graph embedding. Additionally, relevant articles cited in the selected ones underwent the same process to ensure a comprehensive literature review.

Graph representation learning overview

A graph is a powerful data structure to represent entities and their intrinsic relationships in various domains, and biological networks (Fasoulis, Paliouras & Kavraki, 2021; Li, Huang & Zitnik, 2022; Yi et al., 2022) are one of them. Basically, an adjacency matrix can simply represent a graph, but this data form is sparse and high-dimensional which impedes the graph analysis for downstream applications. Graph representation learning (or graph embedding) aims to convert high dimensional graph-structured data into low dimensional dense vectors while preserving intrinsic graph properties (Chen et al., 2020). The technique is to map each node to a vector in another space where the proximity among nodes is preserved. Up to this date, there have been numerous graph representation learning methods developed to generate node embedding, edge embedding, hybrid embedding, or whole graph embedding for an input graph (Chen et al., 2020). These representation features can be applied in various tasks, such as node classification, link prediction, and graph classification (Khoshraftar & An, 2024). In this section, we summarize some major algorithms which are frequently utilized in protein function prediction.

Translational distance methods

TransE (Bordes et al., 2013) is a foundational model in the field of knowledge graph embedding (KGE). Operating on triplets (h,l,t), where h and t represent head and tail nodes connected by an l-typed edge, TransE learns representations of entities and relations by treating the relationship as a translation operation in the embedding space. It ensures that the vector of the tail entity is proximate to the vector of the head entity plus the vector of the relationship. The objective is to make h+l≈t when (h,l,t) holds in the graph, while in false cases, the distance between h+l and t should be greater.

Random walk based methods

DeepWalk (Perozzi, Al-Rfou & Skiena, 2014) employs a random walk mechanism that starts from a target node in the graph and randomly moves to other nodes within a given number of steps, sampling the sequence of nodes visited. DeepWalk treats this random walk sequence as a “sentence” and adopts word2vec’s Skip-gram model (Mikolov et al., 2013), a word embedding technique, to learn a vector representation for each target node. It takes a one-hot encoding vector representing the target word as input and predicts the neighboring nodes within these walks. DeepWalk is particularly useful in protein functional characterization from biological data, such as PPI networks (Alshahrani et al., 2017; Zhang et al., 2019, 2020).

The Node2vec method (Grover & Leskovec, 2016) also consists of two parts, i.e., generating a set of random walks for each node, and training a Skip-gram model based on the target nodes and their random walk sequences. However, node2vec modifies the first part compared to DeepWalk. While DeepWalk samples random walks uniformly, node2vec introduces a sampling strategy with controlled probabilities p and q. A large p encourages global exploration of the graph and avoids returning to the nodes that are already visited, on the other hand, a larger q biases the walk toward local exploration (Khoshraftar & An, 2024). These two parameters allow the method to flexibly mimic breadth-first search (BFS) and depth-first search (DFS), generating random walk paths that can reflect both global and local graph structure.

Deep learning based methods

Graph neural network (GNN) (Kipf & Welling, 2016a) is a type of neural network consisting of multiple GNN layers designed to process graph-structured data based on a message-passing mechanism. In each layer, the current representation for each node is updated by aggregating its neighbors’ embeddings. The process passes through k layers, if k layers are defined. The final embedding of each node is obtained at the kth layer, which aggregates the node’s neighbors that are k hops away from the node (Khoshraftar & An, 2024).

Graph convolutional network (GCN) (Kipf & Welling, 2016a) is a typical GNN, in which adopts the concept of convolutional neural network (CNN) to operate on graph-structured data. GCNs acquire representations that capture both the local structure of a graph and node features. The learning process is made through convolution operations and the iterative aggregation of neighboring node embeddings (Chen et al., 2020). This deep learning architecture is extensively used to infer protein functionalities via PPI and GO graph representation learning (Zhou et al., 2020; You et al., 2021; Li et al., 2022b).

Graph attention network (GAT) (Veličković et al., 2017) is another kind of GNN that applies an attention mechanism to graph data. The attention mechanism calculates the importance score of other nodes around each node, which allows the aggregation phase to determine the influence of neighbors and focus on important parts of the graph. This allows updating the embedding of a particular node by multiplying the information of its neighbors by different weights. In some cases, such as protein structure based networks, not all connections have the same importance, so this approach can be used to dynamically assign weights and remove noise for more effective node embedding learning (Lai & Xu, 2022).

The GraphSAGE (SAmple and aggreGatE) model (Hamilton, Ying & Leskovec, 2017) works in an inductive learning manner, which can handle unseen nodes. It generates node embeddings using a sample-and-aggregate strategy, in which a node’s representation vector at layer kth is updated by concatenating the aggregation of the neighbors’ embeddings with its own embedding from layer k−1th (Khoshraftar & An, 2024). The aggregators used to assemble information from neighboring nodes can be mean, LSTM, and pooling.

Equivariant graph neural networks (EGNN) (Satorras, Hoogeboom & Welling, 2021) represent a specialized class of GNNs designed to learn graph embeddings while preserving crucial symmetry properties like rotation, translation, or permutation invariance. EGNNs process input data consisting of feature vectors and node coordinates, along with edge information, to generate embeddings vectors. The key distinction of EGNNs from conventional GNNs lies in the consideration of relative squared distances between a node’s coordinates and its neighbors during the message-passing operation (Khoshraftar & An, 2024). By ensuring that the learned embedding vectors remain invariant under specific transformations of the input graph, EGNNs demonstrate robustness in scenarios where maintaining symmetries is essential, such as in 3D molecular structure analysis.

Variational graph autoencoder (VGAE) (Kipf & Welling, 2016b) adopts an autoencoder architecture aimed at minimizing the reconstruction loss between input and output data. VGAE employs GCNs as its encoder, which takes an adjacency matrix and a feature matrix as input to generate embedding vectors. Subsequently, the decoder reconstructs the adjacency matrix based on these embeddings through inner product operations.

Graph representation learning for protein function prediction

Graph representation learning has been incrementally utilized to capture the topological relationships inherit in biological systems, such as PPI networks, protein structures, GO graph, proteins and their GO annotations, to eventually improve the accuracy in protein function prediction (You et al., 2021; Lai & Xu, 2022). In this section, we categorize state-of-the-art GO classifiers that utilize graph embedding techniques based on different types of graph data, including PPI network, protein structure, GO graph, and integrated graph. Figure 1 provides an overview, highlighting the primary graph-structured data categories discussed in this review.

Figure 1 An overview of the major graph-structured data used for PFP.

The four key biological graphs highlighted are the PPI network, protein structure, GO graph, and integrated network, each containing relevant subcategories reviewed in this study.

PPI network

PPI networks, where edges represent interactions between protein nodes, are a crucial source of information to assist in silico protein functional annotation. While there are several databases storing PPI networks (Shehu, Barbará & Molloy, 2016), the Search Tool for the Retrieval of Interacting Genes/Proteins (STRING) (Szklarczyk et al., 2023) is the most commonly used for protein function annotation. In the early of 2000s, PPIs were utilized to elucidate protein roles in two schemes, i.e., directly functional propagation on the network, and module assisted ones which first identify network modules and subsequently annotate their functions (Sharan, Ulitsky & Shamir, 2007).

With the recent rise of graph embedding techniques and deep learning, there are two prevalent approaches based on PPI graph embedding to enhance PFP performance, i.e., utilizing PPI network embeddings solely or integrating them with other data sources. The following studies were categorized based on these two research directions, but they differ in terms of detailed model architecture, training, and evaluation datasets. Figure 2 outlines a main workflow of these GO annotation systems, divided into two primary stages: data embedding and prediction module. The blue box indicates whether the method is using PPI network embeddings only or integrating them with sequence based features. Specific components in each methods are described in Table 1, while model evaluations are summarized in Table 2.

Figure 2 A general flowchart of PPI network embedding for PFP, outlining a representative workflow where specific methods may vary in detail.

Dotted arrows indicate optional components. Typically, the PPI network, with or without protein node features derived from subsequences, undergoes a graph representation learning process using methods such as DeepWalk, node2vec, or GCN based model to produce protein node embeddings. For methods that rely solely on PPI network embeddings, these representation vectors are input into a machine learning model or dense layers to predict GO terms. Alternatively, PPI network embeddings can be combined with sequence based features to create concatenated protein representations, which are then used for GO term prediction.

Table 1 PPI graph embedding for PFP (model components).

PFP method	Subcategory1	Data embedding	Prediction module	
Sequence	Subsequence	PPI network	
DeepNF	(1)	–	–	RWR, PPMI, MDA	SVM	
MGEGFP	(1)	–	–	RWR, GCN	LightGBM	
DeepGraphGO	(1)	–	InterProScan	GCN	FC layer	
DeepGO	(2)	Embedding layer, CNN, pooling layer	–	Neuro-symbolic embedding	FC layer, Hierarchical classification	
DeepFunc	(2)	–	InterProScan, FC layer	DeepWalk	FC layer	
SDN2GO2	(2)	ProtVec	Sorting, sparse layer	Trapezoidal NN	Weighted classifier	
DeepGOA	(2)	word2vec, bi-LSTM, CNN	InterProScan, FC	DeepWalk	FC layer	
Prot2GO	(2)	ProtVec, CNN-RNN-Attention	–	node2vec	FC layer	
MultiPredGO3	(2)	Embedding layer, CNN, pooling layer	–	Neuro-symbolic embedding	FC layer, Hierarchical classification	
DeepFusionGO	(2)	ESM-1b	InterPro	GraphSAGE	FC layer	
MSF-PFP	(2)	Global attention and CNN based module	InterPro, CNN based module	MLP	MLP	
Notes:

1 Number represents for subcategories in PPI network embedding for PFP, including (1) PPI network embedding and (2) Integrated PPI network embedding.

2 SDN2GO generates predictions from three sub-models, each processing a different input data type, and then combines them using a weighted classifier to infer GO predictions.

3 MultiPredGO has an additional feature, along with sequence and PPI network embedding: a voxel based 3D structure learned by ResNet-50 and two FC layers.

Table 2 PPI graph embedding for PFP (model evaluation).

PFP method	Subcategory1	Evaluation2		
Test data source	Fmax	DL based competing methods	
MFO	BPO	CCO	
DeepNF	(1)	GOA (Huntley et al., 2015)	*	*	*	–	
MGEGFP	(1)	GO human	*	*	*	DeepNF, DeepMNE-CNN (Peng et al., 2021)	
DeepGraphGO	(1)	Swiss-Prot 2019–2020	0.623	0.327	0.692	DeepGOCNN4, DeepGOPlus	
DeepFunc	(2)	Swiss-Prot 2016	0.56	–	–	DeepGO	
SDN2GO	(2)	GOA 2014–2015	0.471	0.361	0.432	DeepGO	
DeepGOA	(2)	Swiss-Prot 2016	0.558	0.422	0.673	DeepGO	
Prot2GO	(2)	Swiss-Prot human	0.646	0.612	0.718	DeepGOA(b)3, DeepGO, DeepGOPlus4, SDN2GO	
MultiPredGO	(2)	Swiss-Prot	0.3671	0.3278	0.5366	DeepGOSeq, DeepGO	
DeepFusionGO	(2)	DeepGraphGO data	0.632	0.340	0.700	DeepGOPlus, DeepGraphGO	
MSF-PFP	(2)	UniProt human 2024	0.506	0.336	0.493	DCLG (Elhaj-Abdou et al., 2021), DeepGOZero (Kulmanov & Hoehndorf, 2022), PFmulFL (Xia et al., 2022), DeepGOPlus, SDN2GO	
Notes:

1 Number represents for subcategories in PPI network embedding for PFP, including (1) PPI network embedding and (2) Integrated PPI network embedding.

2 As evaluation approaches vary in each study, a part of the evaluation information is included here for a relative comparison. For more details, please refer to each specific method.

3 We add “(b)” to DeepGOA(b) to distinguish between two methods with the same name: one being DeepGOA used PPI network embedding, the other DeepGOA (Zhou et al., 2020, 2019a) used GO graph embedding.

4 DeepGOPlus (Kulmanov & Hoehndorf, 2020) is a successor of DeepGO, a sequence based PFP method combining with DIAMOND sequence similarity scores. DeepGOCNN is DeepGOPlus without DIAMOND scores.

* Scores reported in plots.

PPI network embedding

Herein, researchers focus on learning the protein representations from PPI network using deep learning models. PPI networks can be learned purely or fused with the subsequence information as node features.

In a usual manner, a STRING network represents all of seven active interaction sources, including textmining, experiments, databases, co-expression, neighborhood, gene fusion and co-ocurrence. However, DeepNF (Gligorijević, Barot & Bonneau, 2018) and MGEMFP (Li et al., 2022b) consider PPI in six networks, representing the interaction types above separately, except textmining. For such multiple networks, these two studies both employed an encoder-decoder strategy to reconstruct the data, then the node embeddings extracted from the middle layer in the model will be fed to a classifier, to annotate protein roles. In particular, DeepNF converted six STRING networks for a same set of proteins into matrices by Random Walk with Restarts (RWR) (Tong, Faloutsos & Pan, 2006) at first and then positive pointwise mutual information (PPMI). Subsequently, the matrices are combined and transformed via multimodel deep autoencoder (MDA). The compressed network features are obtained and input to a SVM model to finally deduce GO prediction probabilities. Later, the MGEMFP method considers each interaction network a view, and adopts GCN to jointly learn multiple PPI networks. Initially, features for each individual network are generated by applying RWR method. Then both adjacency matrix representing the network and the initialized features undergo the encoder stage, which is a dual-channel GCN encoder, consisting view-specific channel to disentangle the view-specific information and consensus channel to explore the common pattern across diverse views. Before entering the decoder part, a multi-gate module is established to compute the different contributions of each view in each reconstruction process. Finally, each network representation is extracted from the middle layer and concatenated to pass through a LightGBM model for functionally annotating proteins.

Instead of working on single species separately, DeepGraphGO (You et al., 2021) combined proteins of all species for training only one single model. The InterPro (Paysan-Lafosse et al., 2023) one-hot encoding vectors added as node features is first transformed using a fully connected (FC) layer. Then multi-GCN layers take the input networks to update the high-order representation vector of each protein through their neighboring nodes and edges, followed by an output layer to anticipate GO term scores.

Integrated PPI network embedding

In this approach, PPI based representations are concatenated with other features, such as sequence based, subsequence based, and structural information. The combined vectors are then processed through neural network layers to predict GO terms. The subsequence based features here are usually one-hot encoding vectors representing protein families, domains and motifs fetched from InterPro database.

DeepGO (Kulmanov, Khan & Hoehndorf, 2018) was one of the pioneer deep learning based methods in which combined sequence and PPI features to elucidate GO annotations and showed promising results. The authors encoded protein sequences as lists of amino acid trigrams and learned their features via an embedding, 1D convolution and max pooling layer. Meanwhile, topological representations in PPI networks were extracted by using a neuro-symbolic representation learning method on biological knowledge graphs (Alshahrani et al., 2017). The vectors merged from two types of features were passed through an FC layer and a hierarchical layer to characterize protein functions. In DeepFunc (Zhang et al., 2019), the topological features of PPI network are extracted by DeepWalk. For subsequence features, 35020-dimensional binary vectors from InterPro are passed through FC layers to create dense embeddings. SDN2GO (Cai, Wang & Deng, 2020) integrates three sub-models, each processing a distinct information source: protein sequence, protein domain, and PPI data. Specifically, protein sequences are encoded using an approach similar to DeepGO, but with an increased maximum sequence length of 1,500. The trigram representations are first transformed using ProtVec (Asgari & Mofrad, 2015), then passed through three convolution modules, each consisting of a 1D convolution layer followed by a 1D max-pooling layer. Protein domain is sorted and embeded by a sparse layer. The PPI network represents the interaction information in scored matrix, which is converted into dense features using a trapezoidal neural network module. All three sub-models have an FC and output layer at the end to produce outputs which are integrated together and fed to a weighted classifier to achieve final predictions of GO terms. DeepGOA (Zhang et al., 2020) proposed to improve DeepGO architecture, by integrating protein sequences, subsequence properties and PPI network data. Firstly, sequence based features are learned in both local and global scale via a pipeline of word2vec, a bi-LSTM and a multi-scale CNN layer. Secondly, subsequence one-hot encoding vectors from InterPro are compacted by a dense layer, before concatenated with the sequence embeddings. The topology features in PPI network are learned using the DeepWalk algorithm. Finally, DeepGOA fused all the features together to infer functions for proteins. Prot2GO (Zhang et al., 2021) which was previously presented as GONET (Li et al., 2020) utilized protein sequence and PPI network data. The sequence with maximum 1500-length is mapped to ProtVec and then fed to CNN-RNN-Attention model to generate high-order features. The topological information in the PPI network is captured by node2vec. These two type features are concatenated for function predictions. MultiPredGO (Giri et al., 2020) amalgamates protein sequence, structure, and interaction network data to predict protein functions. Sequential features and PPI representations are extracted using a pipeline similar to DeepGO. Additionally, each 3D proteint structure is converted into four types of voxel based representations and processed through ResNet-50, followed by dense layers to capture relevant structural information. The protein embeddings from these three modalities are then concatenated through a dense layer and fed into a hierarchical classifier to predict GO term probabilities. Another prediction method, DeepFusionGO (Huang, Zheng & Deng, 2022) is designed to mine potential functional clues from fusing the protein sequence, InterPro domain and PPI embeddings. Protein sequence embeddings are generated from pretrained language model ESM-1b (Rives et al., 2021), then fused with embedded InterPro domain binary vectors. DeepFusionGO employs the GraphSAGE algorithm to propagate each protein node feature through the edges in the PPI network and produce PPI based protein embeddings. Eventually, protein representation vectors from GraphSAGE layer and ESM-1b model are concatenated to predict GO label scores for each protein. The MSF-PFP framework (Li et al., 2024b) involves two primary stages: feature extraction and feature fusion, aiming to classify protein functions. In the feature extraction stage, MSF-PFP utilizes token embedding, a global attention module, and CNN layers to capture multi-scale perspectives, encompassing individual, local, and overall sequential features. Similarly, it also examines the local-individual features of protein domains. Concurrently, PPI representations are generated using an MLP module. In the feature fusion stage, all the extracted features are integrated using a block of dense layers to predict GO term scores.

Protein structure

Proteins have a primary structure in the form of an amino acid sequence, yet they fold into 3D conformations to perform a wide range of functions. Thanks to the advances in experimental techniques and in silico structural annotation models, there has been a significant increase in protein structures over the last 20 years (Fasoulis, Paliouras & Kavraki, 2021). Among numerous specialized repositories, the Protein Data Bank (PDB) (Berman et al., 2000) is the most globally recognized database for experimentally-determined 3D protein structures, and the AlphaFold Protein Structure Database (AFPSD) (Varadi et al., 2022) is for predicted ones.

The assumption that proteins with structural resemblance may share similar functions, even if disparate sequences (Krissinel, 2007), together with the availability of structural data, motivate several studies leveraging protein tertiary structures for functional annotation (Fasoulis, Paliouras & Kavraki, 2021). The 3D structure representation based on voxels, as used in MultiPredGO, is not memory efficient because most of the 3D space remains unoccupied by the protein. In this section, 3D atomic coordinates representing protein structures are typically converted into contact maps, which can be effectively learned using geometric deep learning (Bronstein et al., 2017), particularly with GNN architectures. A contact graph Cα−Cα can be constructed as follows, each node is a residue, two resides have contact (connected by an edge) if the distance between their corresponding Cα atoms is within a predefined distance cutoff δ.

A general flowchart of PFP studies using protein structure embedding is shown in Fig. 3. The blue box indicates the methodology for using protein structure: experimental, predicted, or combined protein method. The details of each method are described below, while specific model components compared to the flowchart, along with the evaluation metrics for each method, are summarized in Tables 3 and 4, respectively.

Figure 3 A general flowchart of protein structure embedding for PFP, outlining a representative workflow where specific methods may vary in detail.

Dotted arrows and boxes indicate optional components. In this context, 3D protein structures are represented as contact maps, showing distance contacts between amino acid residues. Initially, a contact graph for each protein, which may include initial node features such as sequence one-hot encoding vectors or per-residue embeddings from a pretrained protein language model, is processed through a module comprising graph embedding techniques such as GCN, GAT, or EGNN. The resulting whole graph embedding captures structural protein features, which can be integrated with the sequence-level embedding to ultimately infer function predictions.

Table 3 Contact graph embedding for PFP (model components).

PFP method	Subcategory1	Data embedding		
Sequence	Contact map	Prediction module	
Residue-level2	Sequence-level		
DeepFRI	(1)	LSTM based pretrained model	–	PDB, GCN	FC layer(s)/MLP	
PersGNN3	(1)	One-hot encoding vectors	–	PDB, GCN		
GAT-GO	(2)	ESM-1b, Sequential features, CNN	ESM-1b	RaptorX, GAT		
Struct2GO	(2)	–	SeqVec	AF2, node2vec, GCN & Self-attention graph pooling		
TransFunc	(2)	ESM-1b	ESM-1b, EGNN	AF2, EGNN		
GPSFun	(2)	ProtT5-XL-U50	–	ESMFold, GNN		
HEAL	(3)	ESM-1b, One-hot encoding vectors	–	PDB/AF2, GCN, HGT		
Notes:

1 Number represents for subcategories in protein structure embedding for PFP, including (1) Experimental protein structure, (2) predicted protein structure and (3) combined protein structure.

2 If the residue-level embeddings are generated only to produce sequence-level vectors and are not used as input for the method, they are not included here.

3 PersGNN extracted an additional feature from protein structure, which is the persistence diagram learned by the persistence network.

Table 4 Contact graph embedding for PFP (model evaluation).

PFP method	Subcategory1	Evaluation2		
Test data source	Fmax	DL based competing methods	
MFO	BPO	CCO	
DeepFRI	(1)	SIFTS 2019	*	*	*	DeepGOSeq3	
PersGNN	(1)	SIFTS	*	*	*	–	
GAT-GO	(2)	DeepFRI data	0.633	0.492	0.547	DeepFRI	
Struct2GO	(2)	EMBL-EBI human	0.701	0.481	0.658	DeepGO, DeepGOA(b), DeepFRI, GAT-GO	
TransFunc	(2)	CAFA3	0.551	0.395	0.659	DeepGO, DeepGOCNN, TALE	
GPSFun	(2)	DeepGraphGO	0.641	0.336	0.726	DeepGOPlus, DeepGraphGO, SPROF-GO (Yuan et al., 2023)	
HEAL	(3)	DeepFRI data	0.747	0.595	0.687	DeepGOSeq, DeepFRI	
Notes:

1 Number represents for subcategories in protein structure embedding for PFP, including (1) experimental protein structure, (2) predicted protein structure and (3) combined protein structure.

2 As evaluation approaches vary in each study, a part of the evaluation information is included here for a relative comparison. For more details, please refer to each specific method.

3 DeepGOSeq is DeepGO trained with protein sequences only.

* Scores reported in plots.

Experimental protein structure

DeepFRI (Gligorijević et al., 2021) is among the pioneering studies that convert 3D structures into contact maps, integrate them with residue embeddings as node features, then apply GNN based models to convolve residue level features over the graph, ultimately constructing protein representations for functional annotation. DeepFRI built an LSTM-based pretrained model using Pfam (Mistry et al., 2021) protein sequences to extract features for each amino acid. Contact maps were constructed from PDB 3D atomic coordinates with distance threshold 10 Å. The model architecture consists of three GCN layers propagating node information over residues in the structure, and two FC layers receiving the final GCN protein representation to produce function predictions. Inspired from DeepFRI, however PersGNN, Swenson et al. (2020) only used one-hot encoding vectors for amino acids, instead of extracting their features from a language model. On the other hand, PersGNN focused on protein structures as the authors stated that structural features may not be fully captured in a contact map. Therefore, contact maps ( δ = 8 Å) embeddings are trained with a GCN model, then incorporated with persistent homology learned by 1D, 2D persistance networks. The final concatenated representations are fed into dense layers to annotate MF GO terms.

Predicted protein structure

Another work inspired by DeepFRI, called GAT-GO (Lai & Xu, 2022), employs the GAT architecture which flexibly aggregates node features through self-attention. Moreover, GAT-GO is one of methods promoting predicted protein structures, herein contact graphs were acquired from RaptorX model (Xu, Mcpartlon & Li, 2021). The residue features were CNN-encoded from sequential features and ESM-1b based residual level embeddings. Then a GAT based block took these graphs and their node features to deduce contact map embeddings. Eventually, GAT-generated and ESM-1b based protein representation vectors are concatenated, to assign GO functions for proteins. On the other hand, Struct2GO (Jiao et al., 2023) distills only AF2 predicted protein structures instead of considering the experimentally-validated ones. Protein sequence representation vectors are extracted by a pretrained protein language model (pLM) SeqVec (Heinzinger et al., 2019). Meanwhile, contact graphs constructed from AF2 3D protein structures firstly undergo node2vec for initial node embeddings, and then two neural network blocks, each comprising a GCN layer, a self-attention graph hierarchical pooling layer, and a readout layer, for final graph embeddings. Sequential and structural features are ultimately fused to produce GO term probabilities. AF2 3D structures are also exclusively utilized in TransFunc (Boadu, Cao & Cheng, 2023). However, instead of using a distance threshold approach to assign an edge between nodes, TransFunc used k-NN approach where each residue is connected to its k nearest neighbors. Following the same paradigm as previous studies, per-residue and per-sequence representation vectors are extracted using ESM-1b. The initial node features in each protein graph include residue-level embeddings and the 3D coordinates of each residue. Those graphs are learned by three consensus EGNN blocks, while the ESM-1b protein sequence embeddings undergo another block, EGNN3. The overall graph features of EGNN1, EGNN2 and EGNN4 are then fused with outputs from EGNN3, to assign GO functions for each protein. GPSFun (Yuan et al., 2024) is a web server that enables users to predict protein functions at both residue and protein levels, including binding sites, GO annotation, subcellular location, and protein solubility. The model architecture underlying the tool operates as follows: For each input sequence, ESMFold (Lin et al., 2023) is used to predict the protein structure, while the pLM ProtT5-XL-U50 (Elnaggar et al., 2022) generates sequence embeddings. A geometric featurizer then constructs a protein-attributed graph, encompassing the residual and relational geometric contexts within the predicted structure. This graph is subsequently input into several GNN layers to produce the final residue representations, which are utilized for downstream tasks such as GO function predictions.

Combined protein structure

Ma et al. (2022) does not propose a novel functional annotation system, however this study creates a benchmark dataset, including PDB-A, PDB-B which are two subsets of DeepFRI dataset, and AF comprising AlphaFold-predicted structures of proteins in PDB-B. Ma et al. (2022) employs the DeepFRI model to assess whether augmenting the training set with additional AlphaFold-computed structures enhances the performance of the function prediction model. They further compare the performance between two models trained exclusively with real structures and computed structures, respectively. Recently, Hierarchical graph transformEr with contrAstive Learning (HEAL) (Gu et al., 2023) is a novel sophisticated architecture to characterize proteins. Input contact maps are queried from both experimental PDB and predicted by AF2 model in AFPSD, while residue-level embeddings are acquired from ESM-1b and one-hot encoding vectors. In addition, the graphs undergo HEAL which comprises a message-passing paradigm GCN encoder and a hierarchical graph transformer (HGT) module. The GCN encoder aims to collect local information in the contact map, which is followed by the HGT which consists of super-nodes generation to explore global graph structure, and attention pooling to pool the super-node representations into graph-level vectors. Also, contrastive learning assists the learning process by perturbing node embeddings to provide different views of the graph representation, and maximizing the similarity between those differences.

GO graph

General GO term embedding

Because of the GO’s complex structure and utilization in many bioinformatics tasks (Smaili, Gao & Hoehndorf, 2018, 2019; Zhong, Kaalia & Rajapakse, 2019; Kim, Kim & Sohn, 2021; Edera, Milone & Stegmayer, 2022), GO graph representation learning, particularly GO term embedding has been studied in numerous approaches. Onto2Vec (Smaili, Gao & Hoehndorf, 2018) represented GO graph edges into axioms and considered these as sentences, for instance, A SubClassOf B. Then word2vec was employed to learn a representation of each ontology. The model also extended to protein annotation and produced protein embeddings as well. OPA2Vec (Smaili, Gao & Hoehndorf, 2019) improved Onto2Vec by incorporating metadata ontology with the Onto2Vec’s corpus. Consequently, the method can produce feature vectors for various biological entities, such as GO, genes and diseases. Later, GO2Vec (Zhong, Kaalia & Rajapakse, 2019) used node2vec to transform GO terms and proteins in a GO annotation graph into feature vectors. Attempting to generate both function and gene representations, however HiG2Vec (Kim, Kim & Sohn, 2021) applied Poincaré embedding whose learning process is in a hyperpolic space, particularly Poincaré ball of GO and Gene Ontology Annotation (GOA) corpus. Recently, anc2vec (Edera, Milone & Stegmayer, 2022) applied neural networks to reconstruct three information types for each GO class, including ontological uniqueness, ancestors hierarchy and sub-ontology membership. The model thus produced optimal learned embeddings for each GO term. GOGCN (Tian et al., 2022) designed a GCN based KGE model comprising an encoder and a decoder. The encoder component consisted of GCN layers that first learned the representations of terms and interactions based on the semantic inter-relationships among GO classes. Subsequently, the decoder stage took the embeddings generated by the encoder to perform secondary graph representation learning through GO term link prediction. GT2Vec (Zhao et al., 2022) learned the representations of GO terms in two phases. Initially, GO names and descriptions were concatenated and fed into a fine-tuned ouBioBERT (Wada et al., 2020) model, producing textual features for GO nodes. The adjacency matrix representing the GO graph, along with the term semantic features obtained in the first stage as node features, were then processed by a GO encoder consisting of graph isomorphism network (GIN) layers. Using a contrastive learning paradigm, the graph representation learning phase ultimately generated the final GO term embeddings.

In the aforementioned studies, the GO node representation vectors were initially generated and then evaluated in several downstream tasks, such as GO semantic similarity, protein functional similarity, PPI, or gene-disease association prediction. However, they were not utilized for protein function prediction.

GO term embedding for PFP

In protein functional annotation, GO representations are typically combined with sequential features to output GO term probabilities in an end-to-end fashion. A general paradigm of these PFP studies is presented in Fig. 4. Additionally, specific model components and evaluation information are summarized in Tables 5 and 6, respectively.

Figure 4 A general flowchart of GO graph embedding for PFP, outlining a representative workflow where specific methods may vary in detail.

Dotted arrows and boxes indicate optional components. Initially, the GO graph, extracted from the GO database, along with additional node features such as GO names and descriptions, is fed into a graph representation learning process based on graph embedding algorithms, such as GCN or PO2Vec, to produce representation vectors for each GO term. Additionally, protein residue-level or sequence-level embeddings can be jointly learned with GO embeddings through various approaches, such as matrix multiplication or cross-attention operation, to emphasize the associations between GO functions and the corresponding proteins. These joint embeddings can then pass through dense layers to ultimately compute the final prediction probabilities.

Table 5 GO graph embedding for PFP (model components).

PFP method	Data embedding	Prediction module	
	Sequence	GO graph		
	Residue-level1	Sequence-level	Node feature	Graph embedding		
DeepGOA(b)	–	One-hot encoding vectors, CNN, FC layer	GO correlations	GCN	Matrix product	
TALE	One-hot encoding vectors, Learnable matrix and positional embedding, Transformer encoder	–	–	Learnable matrix	Matrix product, 1D convolutional and max-pooling layer, FC layer	
DeepChoi	–	SeqVec	GO correlations	GCN	Matrix product	
GCL-GO	–	ESM-1b, FC layer	GO correlations; GO names and definitions	GCN	Matrix product	
PANDA2	–	ESM-1b, FC layer, PAAC	Alignment scores	GNN	GNN	
PFresGO	ProtT5, Autoencoder; One-hot encoding vectors, FC layer	–	–	Anc2vec, Self-attention	Cross-attention, FC layer	
Zhao et al. (2023)	–	ESM-1b, FC layer	–	GCN, Attention	Matrix product	
DeepGATGO	–	ESM-1b, FC layer	GO descriptions	GAT	Matrix product	
PO2GO	–	ESM-1b, MLP	–	PO2Vec, MLP	Matrix product, MLP	
Note:

1 If the residue-level embeddings are computed only to produce sequence-level vectors and are not used as input for the method, they are not included here.

Table 6 GO graph embedding for PFP (model evaluation).

PFP method	Evaluation1	
	Test data source	Fmax	DL based competing methods	
		MFO	BPO	CCO		
DeepGOA(b)	Swiss-Prot human	0.477	0.385	0.629	DeepGO, DeepGOPlus, DEEPred	
TALE	CAFA3	0.548	0.398	0.654	DeepGO, DeepGOCNN, DeepGOPlus, TALE+2	
DeepChoi	CAFA3	0.518	0.470	0.637	DeepGOCNN, TALE	
GCL-GO	CAFA3	0.613	0.516	0.677	DeepGOCNN	
	TALE	0.636	0.384	0.682	DeepGOPlus, TALE, TALE+	
PANDA2	Swiss-Prot 2016	0.598	0.478	0.709	DeepGOPlus, UDSMProt (Strodthoff et al., 2020)	
PFresGO	DeepFRI	0.6917	0.5678	0.6737	DeepGO, DeepFRI, TALE+, DeepGOZero	
Zhao et al. (2023)	Swiss-Prot 2022 human	0.69	0.4895	0.6837	DeepGOCNN, DeepGOA(b), DeepChoi	
DeepGATGO	CAFA3	0.617	0.528	0.679	DeepGO, DeepGOPlus, DeepGraphGO, TALE, GCL-GO	
PO2GO	CAFA3	0.611	0.526	0.648	DeepGOPlus, ESM-1b & DeepGOA3, ESM-1b & TALE3	
Notes:

1 As evaluation approaches vary in each study, a part of the evaluation information is included here for a relative comparison. For more details, please refer to each specific method.

2 TALE+ is TALE combined with DIAMOND score.

3 PO2GO replaced the original protein feature extractor of TALE and DeepGOA with ESM-1b.

DeepGOA(b) (Zhou et al., 2019a, 2020) computes GO correlations based on the information content representing GO hierarchy and the number of proteins annotated with a term. These empirical scores are added as node features into the GO DAG. The graph is then fed into a stack of GCN layers to capture the latent inter-relations and semantic representations of ontology terms. Simultaneously, multi-scale CNN layers extract sequence features from the one-hot encoding of amino acids. A dot product between the learned sequential and GO term features is finally computed to anticipate the association between functions and proteins. TALE (Cao & Shen, 2021) converts the hierarchical information of a GO graph into a one-hot encoding matrix. The GO label matrix is embedded through a trainable look up matrix, while protein sequences are undergone the same approach, along with positional embedding. Subsequently, only the embedded protein matrix is fed through a transformer encoder, as the GO matrix remains fixed for each ontology. The relationships between sequences and labels are jointly learned via matrix products, 1D convolutional and max-pooling layer, before being input to an FC layer to produce GO term probabilities. Considering both sequence and GO learning aspects in DeepGOA(b) and TALE, another method Choi et al. (2021), referred to as DeepChoi in Zhao et al. (2023), proposed a GO annotation system as follows. Instead of using a 1D CNN as in DeepGOA(b), protein features are encoded by SeqVec. Meanwhile, GO graph representation learning in DeepChoi is similar to the approach used in DeepGOA(b), rather than relying solely on the GO hierarchy matrix of one-hot vectors as in TALE. GCL-GO (Choi, Lee & Kim, 2022) models protein sequences using ESM-1b followed by an FC layer. Meanwhile, GO term embeddings are learned from two perspectives. First, the GO graph, with dense one-hot encoding vectors as node features and GO correlations as edge features, is modeled using GCN layers within a contrastive learning framework to produce topological features for each GO term. Second, semantic information is acquired from GO names and definitions, which are embedded using BioBERT (Lee et al., 2020). These semantic and structural features of GO classes are then merged and fused with protein vectors to compute the probabilities of each GO class for the input protein. Another prediction protocol PANDA2 (Zhao, Liu & Wang, 2022) is extended from PANDA (Wang et al., 2018) which is an alignment based protein function prediction system. PANDA2 utilizes three GNN blocks to model the GO DAG with node features, edge features, and global features. Each GO node feature is a vector consisting of 12 alignment results, encompassing the top 10 PSI-BLAST scores, a DIAMOND score, and a priority score. An edge feature denotes if there is a relationship between GO nodes. The global feature is a 20-length vector representing pseudo amino acid composition (PAAC). Additionally, ESM-1b based protein embeddings are merged into GO node features. Eventually, all of those features are concatenated at the third GNN block to output the GO probabilities.

PFresGO (Pan et al., 2023) dynamically focuses on functional amino acids to capture the relationships among key functional regions in protein sequences and GO terms using a multi-head attention mechanism. Specifically, GO vocabulary embeddings are initially generated by anc2vec and then fed into a multi-head attention module. Meanwhile, residue-level representations of each protein chain are obtained using the pretrained pLM ProtT5 (Elnaggar et al., 2022) and compressed by an autoencoder module before being combined with one-hot sequence embeddings. Finally, the correlation between protein features and functions is computed using a cross-attention operation, which receives GO embeddings as a query to detect related protein information. This is followed by FC layers to produce the final function predictions. Zhao et al. (2023) aimed to further enhance the GO DAG learning approach used in DeepGOA(b) and DeepChoi. First, two separate GO graphs representing topological and functional information are created. The authors explained that these two types of information infer distinct GO knowledge, therefore a mixed graph like the one in the former studies might introduce noise into the prediction system. Subsequently, multi-view GCN layers are applied to adaptively learn the functional information, topological structures, and their combinations. For protein representation learning, sequence embeddings are extracted using the ESM-1b model. Then, sequential and functional features are fused using a dot product to achieve GO prediction scores. DeepGATGO (Li, Jiang & Li, 2023) recently introduced novel suggestions to characterize the link between protein sequences and GO functions. Residue-level vectors are obtained from the ESM-1b model followed by a dense layer, as is common practice. However, GO graph embedding is considered from two perspectives: topological and semantic information. The structural information is learned by a GAT model, which uses a GO one-hot encoding matrix as node features and an adjacency matrix as edge features. Meanwhile, GO descriptions, such as names and definitions, are embedded using the pretrained biomedical language model BioBERT. These semantic representation vectors are used for data augmentation to create negative and positive GO samples as input for a contrastive learning module. The learned features of GO classes from both types are merged into the final GO feature matrix, which is then multiplied by the protein sequence matrix to ultimately classify protein-function relationships. In Li et al. (2024a), the authors introduce a novel GO class representation learning method called PO2Vec, which uses a contrastive learning framework based on partial order constraints derived from the shortest reachable path (SRP). The SRP between two terms in a GO graph is defined as the minimum number of edges connecting them, encompassing three possible scenarios: direct reachability, indirect reachability, and unreachability. These constraints guide the embedding process to ensure that the smaller the SRP between two terms, the higher their semantic similarity. Furthermore, the authors propose a function annotation system named PO2GO, which combines PO2Vec with protein sequence embeddings from ESM-1b in a joint modeling predictor to infer GO functions for proteins. Since PO2Vec and PO2GO operate independently, the GO class embeddings generated by PO2Vec can be applied to a variety of biological tasks at both the GO and protein levels.

Integrated graph

Graph data types are not just used individually; researchers also exploit multiple types of networks, or heterogeneous graphs containing different types of nodes and edges, to interpret GO functions for proteins. Since the approaches of PFP systems in this category vary, their details are described below, with model summaries presented in Table 7.

Table 7 Integrated graph embedding for PFP.

PFP method	Subcategory1	Graph embedding	Evaluation2	
			Test data source	Fmax	DL based competing methods	
				MFO	BPO	CCO		
Graph2GO	(1)	VGAE	Swiss-Prot human 2018	0.718	0.490	0.686	DeepNF	
PSPGO	(1)	GAT	CAFA4	0.719	0.388	0.745	DeepGOCNN, DeepGOPlus, TALE, TALE+, DeepGraphGO	
HNetGO	(1)	GNN & Attention	UniProt human	0.697	0.561	0.748	GONET, DeepGO, DeepGOPlus, DeepGOA(b)	
DeepPFP-CO	(1)	GCN	CAFA3	0.570	0.503	0.616	DeepGO, DeepGOA, DeepGOPlus	
PFP-GMB	(1)	GCN	UniProt 2022	0.624	0.453	0.652	DeepGO, DeepGOA, DeepGOCNN, DeepGOZero, DeepGrapGO, ATGO (Zhu et al., 2022)	
OntoProtein	(2)	TransE	GOA	–	–	–	ProtBert	
LATTE2GO	(2)	GAT	DeepGraphGO	0.840	0.574	0.683	DeepGOCNN, DeepGraphGO	
DeepHGAT	(2)	GAT	GOA human	0.8058	0.7928	0.8383	DeepFRI, PANDA2, TALE+, PFresGo, DeepGraphGO, PSPGO	
TALE-cmap	(3)	1D CNN & TALE	TALE	0.648	0.399	0.666	DeepGOPlus, TALE	
SLPFA	(3)	Learnable matrix & SMG	DeepFRI	0.604	0.478	0.524	DeepGO, DeepFRI, GAT-GO	
GNN3DGO	(3)	GAT & GCN	DeepFRI	0.662	0.499	0.552	DeepGO, DeepFRI, GAT-GO	
POLAT	(3)	Learnable matrix & SMG	DeepFRI	0.670	0.515	0.578	DeepGO, GAT-GO	
			TALE	0.663	0.403	0.678	DeepGOPlus, TALE, TALE+, TALE-cmap	
Notes:

1 Number represents for subcategories in integrated graph embedding for PFP, including (1) PPI network and other biological graph, (2) heterogeneous network and (3) protein structure and GO graph.

2 As evaluation approaches vary in each study, a part of the evaluation information is included here for a relative comparison. For more details, please refer to each specific method.

PPI network and other biological graph

There is a group of GO annotation systems applying graph embedding on PPI data and other types of network to enhance the performance of PFP. For instance, Graph2GO (Fan, Guan & Zhang, 2020) is claimed to be the first method that uses both an attributed PPI graph and a sequence similarity network (SSN) to assign GO functions to proteins. Specifically, the SSN is defined from alignment scores (E-values) using the BLAST program. The protein node features are a concatenation of three components: a conjoint triad based sequence representation vector, a subcellular location vector, and a protein domain binary vector. These two networks are learned separately, each using a VGAE model to simultaneously map topological information and node features into the embedding space. The two types of compressed protein embeddings obtained from the VGAE models are then input into a deep neural network to classify GO labels. In another study (Li et al., 2022a), researchers observed that the co-occurrence between ontology terms can improve the prediction of protein roles. This feature is defined as the conditional probability of function GOj occurring when function GOi appears in the training dataset. Consequently, they designed an automatic function prediction method called DeepPFP-CO, consisting of two components: a feature combination part and a function prediction part. The feature combination part follows the general workflow of PPI network embedding for PFP to produce a base prediction of GO functions. The function prediction section consists of two GCN layers, which receive the base prediction and a correlation matrix representing the co-occurrence of GO terms, to further refine the final prediction scores. Orthologous relations from the EggNOG database (Huerta-Cepas et al., 2019) have been partially integrated into the PPI networks used in DeepFunc, DeepGOA, MultiPredGO, and DeepPFP-CO. In a recent study, PFP-GMB (Shuai et al., 2023) network data is comprehensively utilized alongside PPI embeddings to generate functional annotations for proteins. Specifically, protein sequence embeddings are generated using ESM-1b followed by a dense layer, while subsequence InterPro vectors are compressed by an embedding layer to serve as node features in the PPI and orthology networks. For each network, two GCN layers are applied to extract related features. Subsequently, protein embeddings output from the two networks are unified through an attention mechanism and concatenated with the ESM-1b based representations to identify protein-function associations.

Heterogeneous network

On another front, researchers attempt to construct heterogeneous networks for inferring protein characteristics. The authors of PSPGO (Wu et al., 2022) leverage PPI and DIAMOND based SSN to built a cross-species heterogeneous network that includes both protein interactions and sequence similarities. The binary protein node features, derived from InterPro, are first transformed by an embedding layer, followed by an MLP. To mitigate the impact of possible noise in network data, PSPGO uses the GAT architecture to guide message propagation for graph representation learning. The GAT based propagation stage takes the GO label matrix and the unified network with transformed node features, dynamically adjusts the edge weights, and outputs protein features as well as GO label propagation. These learned protein embeddings are fed into an FC layer, which is then weighted combined with the label propagation results to infer GO term probabilities. Another study named HNetGO (Zhang et al., 2023b) introduces a unified graph consisting of two kinds of nodes, proteins and GO terms, and four edge types: interact_with, similar_with, is_a, and annotate. These edges encode protein interactions, sequence similarities between proteins, hierarchical structures between GO classes, and functional associations between proteins and GO nodes, respectively. Protein node features are extracted from SeqVec, while sequence similarity scores are computed from DIAMOND results. HNetGO models the protein-GO term relationships as a link prediction problem, comprising two main steps. First, an attention based GNN includes a node-level mutual attention layer to determine the attention weights for the current node’s direct neighbors, and a multi-head messaging layer to propagate information to the current node based on attention weights with different neighbor types. Second, the protein and GO term outputs from the graph embedding step are fed into a link prediction layer to ultimately elucidate GO terms’ confidence scores. OntoProtein (Zhang et al., 2022) is presented as the first protein pretraining language framework that incorporates a knowledge graph. The knowledge graph, named ProteinKG25, is constructed from the GO DAG and protein annotations released in 2020. Term node features include GO name and description embeddings acquired from BiomedBERT (Gu et al., 2021), while protein node features are sequence embeddings from ProtBert (Elnaggar et al., 2022). OntoProtein jointly learns protein representations using masked protein modeling initialized with ProtBert and knowledge graph representations using TransE. The resulting knowledge-enhanced protein embeddings are utilized in various downstream protein tasks, particularly protein function prediction. LATTE2GO (Layer-stacked ATTention Embedding to Gene Ontology) (Tran & Gao, 2023) extends a complex heterogeneous network encompassing proteins, RNA (mRNA, miRNA, lncRNA), and GO terms, connecting by various physical interactions, genetic interactions, and functional relationships. Since complex biological events usually involve multi-omics entities, integrating these interactions could reveal vital associations for identifying protein functions (Monti et al., 2019). Initial protein node features are also acquired from InterPro. The model stacks GAT layers to learn the significant relations within the network, producing embedding vectors for proteins and GO terms. These feature types are then fused using DistMult (Yang et al., 2014) to compute GO classification scores for each protein. The authors of DeepHGAT (Zhao et al., 2024) observed that most current deep learning based PFP methods consider undefined annotations as negative ones and do not utilize the available negative protein-term associations. To address this, they construct a heterogeneous graph incorporating PPI, GO DAG, and both positive and negative annotation information. Initial protein features are obtained from ESM-1b, while GO terms are represented with one-hot encoding vectors. A GAT based model is then applied to embed the network and output protein and GO node embeddings. These final representation vectors are integrated to predict unobserved positive protein-term links.

Protein structure and GO graph

There is another research direction which focuses on simultaneously learning protein structure and GO graphs to enhance the predictive performance of protein function. TALE-cmap (Qiu, Wu & Shao, 2022) improves the prediction accuracy of TALE by integrating topological structures of proteins. Concretely, protein contact maps are first predicted using the MSA Transformer (Rao et al., 2021), embedded by a 1D CNN layer, and then incorporated into the TALE architecture by concatenating them with the sequence embeddings. SLPFA (Zhang et al., 2023a) implements a framework for structure-GO representation learning to enhance automatic protein function annotation. Initially, residue-level and sequence-level vectors are generated by ESM-1b. Protein contact maps are predicted using RaptorX to represent structural information, with the residue embeddings serving as node features. Meanwhile, the hierarchical information of the GO graph is represented in an adjacency matrix. Next, SLPFA employs a soft-mask GNN (SMG) and a trainable matrix to extract semantics from the structure graphs and hierarchical GO labels, respectively. A dot product with an attention mechanism fuses the two learned matrices into a joint space to bridge the semantic gap between them. Subsequently, the model utilizes a CNN on the joint matrix to compute anchor sequences for proteins and labels separately, facilitating high-level representation learning of proteins and labels. Finally, the two types of representation vectors are concatenated with ESM-1b based sequence embeddings to interpret GO annotations. GNNGO3D (Zhang, Jiang & Yang, 2024) proposes a GO predictor for proteins that combines protein structure and functional hierarchy learning as follows. Protein contact maps are derived from RaptorX, with amino acid node features represented by the position-specific score matrix (PSSM) embedded using a 1D CNN layer. The contact graph representations are firstly obtained by a stack of four GAT layers, each followed by a self-attentional graph pooling (SAGPool) layer, before being merged with protein sequence embeddings from ESM-1b. Additionally, GNNGO3D employs a GCN to capture the topological features for each GO term. Finally, the output protein and functional features undergo a dot product for feature fusion, before being fed into an MLP to classify GO terms. Recently, POLAT (Liu et al., 2024) also acquires residue node features from a pretrained pLM and transforms them using a 1D CNN layer. The model generates representations for the GO adjacency matrix and the predicted contact maps using a learnable matrix and an SGM architecture to adaptively extract relevant residues for the functions, respectively. These two types of embeddings are aggregated by a residue-label attention module to create protein features, which are then concatenated with the pLM based sequence features. The final merged protein representations undergo a fully connected layer to score GO term probabilities. In addition, the choice of pLM and contact map prediction method in POLAT is flexible and depends on the comparison with other PFP methods. Specifically, ESM-1b and RaptorX are used when POLAT compares the model performance among structured based methods, while ESM-2 (Lin et al., 2022) and AF2 are used when comparing with sequence based and multimodal predictors.

Conclusion

Automatic protein function prediction remains an active research area, assisting bioinformatics experts in understanding protein functions. Graph representation learning makes significant contributions to this field by dissecting PPI networks, protein structures, GO graphs, protein-GO annotations, and multi-omics networks. While GNNs are the most frequently used architecture, autoencoders, neural networks, and hybrid approaches are also employed to provide deeper insights into the functional landscape of proteins.

The increasing availability of biological graphs, coupled with advancements in graph embedding models, has led to the development of numerous graph embedding based protein function annotation systems. However, this research area still faces several challenges. In terms of data, incomplete information is a prevalent problem in PPI networks, protein structures, and GO annotations. Specifically, PPI networks are affected by noise due to false positive interactions. Reviewed functional and structural annotations of proteins are limited because the experimental process is expensive and time-consuming, and the data is often imbalanced resulting from numerous factors. Additionally, the GO tree is complex, with many types of relationships, leading most studies to focus on the major ones. Regarding graph learning models, most GNNs suffer from the over-smoothing problem when stacking multiple layers, a challenge that remains unresolved.

The research ground opens up a huge space for varying future directions, but addressing current challenges in data and model construction is essential. Effective data integration, including the use of heterogeneous graphs and the combination of multiple networks, can leverage each other’s strengths and mitigate noise. The integration of protein sequence data and other features holds promise for accurately inferring protein characteristics. Model and hyperparameter tuning are crucial steps in constructing deep learning based predictors. For instance, several strategies have been suggested to address the over-smoothing issue in GNNs, such as adding self-attention mechanisms and applying regularization techniques. However, the optimal choice depends on the specific data scenario to ensure the best performance while maintaining model generalization. Additionally, developing a dynamic predictive system with proper data organization is vital to easily scale up with large-scale and rapidly changing input resources. Furthermore, explainable learned features are essential to elucidate the decision-making process, especially for researchers without programming expertise.

Additional Information and Declarations

Competing Interests

Author Contributions

Data Availability

The authors declare that they have no competing interests.

Thi Thuy Duong Vu conceived and designed the experiments, performed the experiments, analyzed the data, prepared figures and/or tables, authored or reviewed drafts of the article, and approved the final draft.

Jeongho Kim conceived and designed the experiments, analyzed the data, prepared figures and/or tables, and approved the final draft.

Jaehee Jung conceived and designed the experiments, authored or reviewed drafts of the article, and approved the final draft.

The following information was supplied regarding data availability:

This is a literature review.

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
