# Peer review of "An experimental analysis of graph representation learning for Gene Ontology based protein function prediction"

_PeerJ, doi:10.7717/peerj.18509_

## Round 0.1 · original submission · Major Revisions

Please address all the queries and concerns of the reviewers and amend manuscript accordingly.

Reviewer 1 ·

Basic reporting

The protein function prediction is widely used in fields such as disease research and drug development, and it is necessary to predict with computer models as protein sequences increase. And this paper reviews recently published papers on predicting protein function using deep learning using graphs.

Experimental design

1. While the primary goal of this paper is to summarize the literature on protein function prediction. Still, since the chapter after the “INTRODUCTION” is about “GRAPH REPRESENTATION LEARNING OVERVIEW”, followed by a review of recent papers on “GRAPH REPRESENTATION LEARNING FOR PROTEIN FUNCTION PREDICTION”, it is equally important to provide a comprehensive understanding of the concept of protein function. Therefore, a chapter that delves into the general meaning of protein function is essential.

2. In the ‘GRAPH REPRESENTATION LEARNING FOR PROTEIN FUNCTION PREDICTION section’, the paper discusses four key subsections: PPI Network, Protein structure, GO graph, and Integrated graphs. However, to provide a clear roadmap for the reader, it is crucial to include a mind map-like figure that centers on the keywords being reviewed.

Validity of the findings

3. Figures 1 to 3 are essential because they represent the paradigm for each method. However, the manuscript lacks explanation, and the figures need to be fleshed out a bit more. In particular, Fig 2 does not show how graph embedding is being utilised.
We recommend that the paradigm illustrations in Fig 1 ~ Fig 3 should be more specific.
In Fig. 1, the Data Embedding section is titled ‘Case 2) Use only PPI embedding’, but there are confusing terms such as ‘PPI network embedding only’ in the subtitle and ‘PPI embedding’ in Table 2, so it is necessary to unify them.
In Fig 2, there needs to be an explanation of what the dotted solid line means and why it is a ‘common paradigm of protein structure’. In Fig. 1, the PPI is graphically represented, but there is no explanation of how the graph is used in this structure section. It may be helpful to illustrate the flow of the reviewed papers in the figure by dividing the lines.

4. Tables 1 and 2, which contain the contents of PPI Network, need a column containing the subsection's contents, and Tables 3 through 7, which are included in other Protein structure, GO graph, and Integrated graphs sections, need to be reorganized into tables where the subsection's contents can be organized.

Additional comments

5. Please read the citations carefully and ask for corrections if there need to be more parentheses.

6. The authors must rectify writing errors, including syntax, word spacing, and the conciseness of table and figure titles.

7. The following recently published papers (after 2024 year) utilize garph for gene function prediction. Please refer to them to enrich your review.
https://www.nature.com/articles/s41598-022-12201-9
https://www.sciencedirect.com/science/article/abs/pii/S1476927124000525

Reviewer 2 ·

Basic reporting

1. Is the review of broad and cross-disciplinary interest and within the scope of the journal?

Yes, the review is of broad and cross-disciplinary interest and fits well within the scope of journals focused on bioinformatics, computational biology, and systems biology. The manuscript addresses the intersection of protein function prediction—a fundamental topic in bioinformatics—and advanced computational techniques like graph representation learning and deep learning.



2. Has the field been reviewed recently? If so, is there a good reason for this review (different point of view, accessible to a different audience, etc.)?

The specific focus on graph representation learning techniques applied to Gene Ontology (GO)-based protein function prediction is relatively new and not extensively covered in existing literature. The field has seen rapid advancements recently, particularly with the emergence of powerful graph neural networks and the increased availability of biological data such as AlphaFold 2. This review provides a timely and comprehensive analysis of how graph representation learning is transforming protein function prediction. It offers a fresh perspective by categorizing the methods based on different types of graph data (PPI networks, protein structures, GO graphs, and integrated graphs), which helps readers understand the landscape of current approaches. The review also discusses existing limitations and suggests future research directions, making it a valuable resource for both new and experienced researchers in the field.


3. Does the Introduction adequately introduce the subject and make it clear who the audience is/what the motivation is?

Yes, the Introduction effectively introduces the subject and clearly outlines the motivation behind the review. The target audience is implicitly defined as researchers and practitioners in bioinformatics, computational biology, and related fields who are interested in applying advanced computational techniques to biological problems. The motivation is clear: to bridge the gap between the rapid generation of protein sequence data and the slower pace of manual annotation by leveraging graph-based computational methods.

Experimental design

1. Is the Survey Methodology consistent with a comprehensive, unbiased coverage of the subject? If not, what is missing?

The Survey Methodology section outlines a systematic approach to literature selection, aiming for a comprehensive and unbiased coverage of the subject. The authors focused on articles published after 2019 to emphasize recent advancements. While this helps in highlighting the latest developments, it may inadvertently exclude foundational studies published before 2019 that are essential for understanding the evolution of graph representation learning in protein function prediction.

2. Are sources adequately cited? Quoted or paraphrased as appropriate?
The manuscript references numerous studies throughout the text, indicating that the authors have engaged with a broad range of sources.

3. Is the review organized logically into coherent paragraphs/subsections?
Yes, the review is organized logically into coherent paragraphs and subsections, enhancing readability and comprehension. Including more transition sentences between sections and paragraphs could enhance the flow and connection between different topics.

Validity of the findings

1. Is there a well developed and supported argument that meets the goals set out in the Introduction?
The manuscript presents a well-developed and supported argument that meets the goals outlined in the Introduction. The authors effectively bridge the gap between the initial motivation and the detailed analysis provided, offering a coherent and comprehensive review of the application of graph representation learning in protein function prediction.


2. Does the Conclusion identify unresolved questions / gaps / future directions?
Yes, the Conclusion identifies unresolved questions, gaps, and future directions in the field. The authors provide a thoughtful analysis of current limitations and propose actionable solutions and areas for further research.

Reviewer 3 ·

Basic reporting

The authors introduced four key graph representation methods for predicting protein function: PPI networks, protein structures, Gene Ontology (GO), and integrated graphs. Each method is thoroughly summarized, and the authors clearly present the specific results, offering detailed insights into their application and effectiveness. The paper systematically covers the strengths and limitations of each approach, helping readers understand the distinct roles these methods play in protein function prediction. Overall, the paper is well-organized, informative, and provides a comprehensive overview of the current state of research in this field. The clarity and depth of explanation make it a valuable contribution to the literature.

Experimental design

The study aims to develop a robust and explainable protein function prediction system that effectively addresses the current limitations in data quality, model construction, and interpretability. By focusing on these critical areas, the research seeks to enhance both prediction accuracy and scalability for practical applications in the field. The authors present graph representation learning techniques specifically tailored for protein function prediction, demonstrating their potential to improve outcomes in this area.

In their work, the authors provide a detailed flowchart illustrating the common processes involved in the Gene Ontology (GO) annotation system, as well as model evaluations for each method employed. This visual representation aids in understanding how different approaches contribute to the overall goal of protein function prediction.

The study design is strategically oriented toward tackling the key challenges identified in the literature, including issues related to data quality, the intricacies of model construction, and the necessity for explainable predictions. By outlining a comprehensive approach that integrates multiple biological networks and employs cutting-edge methods in graph representation learning, the authors set a strong foundation for advancing research in this vital area of bioinformatics. Overall, this work not only addresses existing gaps but also paves the way for future developments in protein function prediction systems.

Validity of the findings

The validity of the findings in the study design focused on protein function prediction using graph representation learning (GRL) is contingent upon several critical factors that collectively ensure both internal and external validity. Internal validity is primarily influenced by various aspects such as the quality of the data used, the preprocessing steps undertaken, the design of the model, and how effectively it is implemented. Additionally, the choice of appropriate evaluation metrics plays a significant role in establishing the internal validity of the study, as these metrics provide essential insights into the model's performance and reliability.

On the other hand, external validity pertains to the model's ability to generalize its findings to other datasets outside of those used in the study. This includes considerations of how scalable the model is and how well it can adapt to new, unseen data that may present different characteristics or complexities. While the study demonstrates strong internal validity through careful data handling and rigorous model evaluation, it is essential for the authors to also address the aspect of external validity. This is particularly important in relation to the model’s capability to generalize its predictions to various datasets, ensuring that the findings are applicable across different biological contexts and not limited to a specific set of data. By explicitly discussing external validity, the authors can strengthen their conclusions and enhance the overall impact of their research in the field of protein function prediction.

Additional comments

The writing in this work is generally good, but the review identified several areas for improvement. For instance, in line #302, it should say, "They further 'compare' the performance," correcting the verb tense.

---

## Round 0.2 · accepted · Accept

All concerns of the reviewers are adequately addressed, and the revised manuscript is acceptable now.

Reviewer 1 ·

Basic reporting

The author seems to have done a good job of making all the required modifications. There are no further revision requirements.

Experimental design

no comment

Validity of the findings

no comment

Additional comments

no comment

Reviewer 2 ·

Basic reporting

As previous

Experimental design

As previous

Validity of the findings

As previous

Reviewer 3 ·

Basic reporting

N/A

Experimental design

N/A

Validity of the findings

N/A

Additional comments

The authors have thoughtfully addressed my comments and have made the necessary updates to the manuscript to reflect those changes.